# EMBridge: Enhancing Gesture Generalization from EMG Signals Through Cross-modal Representation Learning

**Wenhui Cui**[1,2]*, **Christopher M. Sandino**[1], **Hadi Pouransari**[1], **Ran Liu**[1], **Juri Minxha**[1],
**Ellen L. Zippi**[1], **Erdrin Azemi**[1], **Behrooz Mahasseni**[1]
[1] Apple
[2] University of Southern California
{wcui23, csandino, mpouransari, ran_liu3, j_minxha, ezippi, erdrin, bmahasseni}@apple.com

## ABSTRACT

Hand gesture classification using high-quality structured data such as videos, images, and hand skeletons is a well-explored problem in computer vision. Alternatively, leveraging low-power, cost-effective bio-signals, e.g., surface electromyography (sEMG), allows for continuous gesture prediction on wearable devices. In this work, we aim to enhance EMG representation quality by aligning it with embeddings obtained from structured, high-quality modalities that provide richer semantic guidance, ultimately enabling zero-shot gesture generalization. Specifically, we propose EMBridge, a cross-modal representation learning framework that bridges the modality gap between EMG and pose. EMBridge learns high-quality EMG representations by introducing a Querying Transformer (Q-Former), a masked pose reconstruction loss, and a community-aware soft contrastive learning objective that aligns the relative geometry of the embedding spaces. We evaluate EMBridge on both in-distribution and unseen gesture classification tasks and demonstrate consistent performance gains over all baselines. To the best of our knowledge, EMBridge is the first cross-modal representation learning framework to achieve zero-shot gesture classification from wearable EMG signals, showing potential toward real-world gesture recognition on wearable devices.

## 1 INTRODUCTION

Hand gesture recognition on wearable devices has recently attracted significant interest (Pyun et al., 2024; Moin et al., 2021) and demonstrated potential across diverse applications such as rehabilitation (Marcos-Antón et al., 2023), human–computer interaction (Jarque-Bou et al., 2021), and prosthetic control (Yu et al., 2023). With advances in deep learning and the availability of large-scale visual data, including videos and motion capture (Casile et al., 2023), vision-based models have achieved remarkable success (Pavlakos et al., 2023; Qi et al., 2024). However, cameras suffer from high power demands and privacy concerns, and potential occlusions can destabilize vision-based classification. This has motivated growing interest in low-power, easily integrable sensors (Tchantchane et al., 2023), such as surface electromyography (sEMG), for gesture recognition on wearable devices (Tchantchane et al., 2023; Wang et al., 2023). Deep learning approaches have been explored for EMG-based gesture classification, including convolutional neural networks (CNNs) (Atzori et al., 2016), recurrent neural networks (RNNs) (Liu et al., 2021), and Transformers (Montazerin et al., 2023). However, predicting hand gestures from wearable EMG, especially generalizing to unseen gestures without task-specific training, remains challenging (Laput & Harrison, 2019). This is mainly because of the high variability and fine dexterity of human hand movements, sensor noise, and/or the limited scale of publicly available data (Lee et al., 2024a; Pereira et al., 2024; Tam et al., 2024). Due to the noisy and heterogeneous nature of EMG signals, learning from EMG alone (through self-supervised learning or supervised end-to-end training) may not reliably yield generalizable and discriminative representations (as later demonstrated in our experiments).

---

*Work done during internship at Apple.

An effective strategy to overcome the above limitations is to leverage another modality that offers richer semantic structure and higher signal quality as guidance during representation learning. This can be achieved through cross-modal representation learning, which has proven highly effective in improving the quality of learned embeddings and shown remarkable success in vision-language models (Xie et al., 2025; Li et al., 2023), audio-visual language models (Gurram et al., 2022; Guo et al., 2025), and in biosignals such as IMU-video-text alignment (Moon et al., 2022) and EEG-text alignment (Feng et al., 2023). However, aligning representations across modalities has not yet been fully explored for wearable EMG signals. Given paired EMG recordings and kinematic hand pose annotations collected simultaneously, we can study cross-modal alignment between EMG and pose. Unlike visual data, pose data directly captures the kinematics of hand movements, making it an informative modality for guiding EMG representation learning. Therefore, we introduce a cross-modal framework for EMG representation learning that leverages a high-quality anchor modality, where pose data provides richer supervisory signals by capturing structural and semantic relationships. Our goal is to improve the quality of EMG embeddings, enabling generalization to new users and unseen gestures at test time without the need for additional training or large-scale data collection. A potential practical application of our framework is wearable Human-Computer Interaction. In scenarios like VR/AR and prosthetic control applications (Jarque-Bou et al., 2021; Yu et al., 2023), a wrist-worn device must continuously infer hand gestures from EMG to drive a virtual avatar or robotic hand. A critical bottleneck is that users cannot be expected to record training data for every possible gesture they might perform. Our framework is designed to enable zero-shot generalization, allowing the model to recognize novel gestures without requiring training samples from users.

We first introduce two unimodal encoders trained separately on EMG and pose data, and then align their output embeddings. Unlike classical approaches such as CLIP or BLIP (Radford et al., 2021; Li et al., 2023; 2022), which symmetrically update both encoders toward a shared latent space, our design adopts an asymmetric setup, where the pose encoder is frozen as an anchor and only the EMG encoder is optimized. On this basis, we propose **EMBridge**, a cross-modal representation learning framework that bridges the modality gap between EMG and pose and enhances the representation quality learned from EMG signals through advanced alignment with pose representations. EMBridge consists of three components: a Querying Transformer (Q-former) (Li et al., 2023) that extracts pose-informative queries and aligns EMG and pose, a masked pose reconstruction loss (MPRL) that encourages queries to carry structured pose information, and a community-aware soft contrastive learning (CASCLe) objective that considers the neighborhood structures of poses and aligns the relative geometry in the latent space across modalities. Standard contrastive learning approaches (e.g., InfoNCE) treat all non-matching samples as equally distant negatives. However, this assumption is suboptimal for our pose data, which is inherently continuous. Moreover, poses across different gesture categories can be semantically close. To capture these structural similarities and avoid confusing the model with hard negatives, we introduce CASCLe, which utilizes geometric proximity in the pose space to generate soft targets. Together, these objectives guide the EMG encoder to capture pose-relevant semantics and produce discriminative, generalizable embeddings. Unlike general-purpose multi-modal alignment, EMBridge is designed as a specialized solution for EMG-based gesture classification through cross-modal supervision.

We utilize large-scale public EMG datasets (Salter et al., 2024; Atzori et al., 2014), which provide simultaneous paired EMG and pose recordings to pre-train our EMBridge model and perform downstream evaluations. We design a gesture classification task to evaluate EMG representation quality. Following CLIP evaluation protocol (Radford et al., 2021), we validate the learned EMG representations through zero-shot classification and linear probing, demonstrating superior performance on both in-distribution and unseen gestures compared to benchmark models. In summary, our contributions are two-fold: (i). We propose a cross-modal representation learning strategy to enhance the quality of EMG representations learned from noisy EMG signals by aligning them with high-quality pose representations. (ii). To the best of our knowledge, EMBridge is the first cross-modal framework enabling zero-shot classification of unseen gestures for EMG signals from wearable devices.

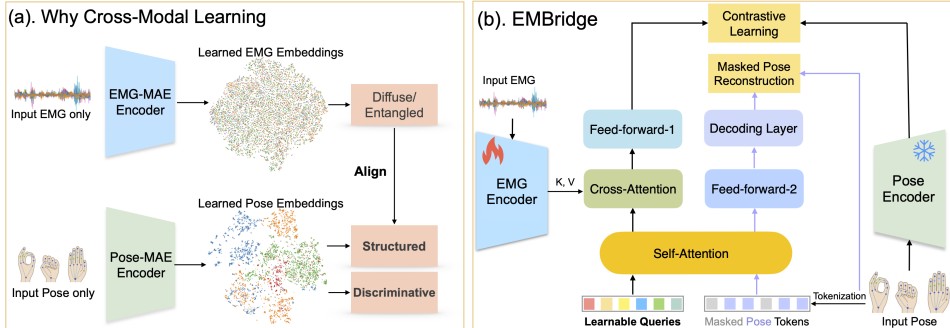

Figure 1: (a) Motivation for cross-modal representation learning: using the same MAE pre-training, pose embeddings are semantically structured and well-separated across gestures (colors), whereas EMG embeddings are not. This motivates leveraging pose as guidance to structure the EMG representation space. (b) Detailed architecture of EMBridge. Only one transformer block (self-attention, cross-attention, and feed-forward layers) is shown for clarity, the model uses four such blocks.

## 2 METHODOLOGY

### 2.1 PRELIMINARIES

***Definition 1. EMG, pose data, and gesture classes.*** Let $\mathcal{X} = \{\mathbf{x}_i \in \mathbb{R}^{C \times T}\}_{i=1}^N$ denote multi-channel EMG sequences with $C$ channels and window length $T$, and let $\mathcal{P} = \{\mathbf{p}_i \in \mathbb{R}^{J \times T}\}_{i=1}^N$ denote the paired pose sequences of joint angles, where $J$ is the number of joints in a predefined hand skeleton (Salter et al., 2024). Here, $N$ is the total number of paired samples. We define a pose as the instantaneous state of the hand skeleton (20 joint angles) at a single time point, whereas a gesture is a temporal sequence of poses over a time window. Let $\mathcal{Y} = \{1, \ldots, K\}$ denote the gesture classes, and let $\{y_i\}_{i=1}^N$ be the labels with $y_i \in \mathcal{Y}$. Each pose $\mathbf{p}_i$ has a unique label $y_i$, although multiple samples may share the same label. The paired dataset is

$$\mathcal{D} = \{(\mathbf{x}_i, \mathbf{p}_i, y_i)\}_{i=1}^N,$$

which we split into $\mathcal{D}_{\mathrm{tr}}$, $\mathcal{D}_{\mathrm{val}}$, and $\mathcal{D}_{\mathrm{test}}$. Let $\mathcal{Y}_{\mathrm{in}} \subset \mathcal{Y}$ be the in-distribution gesture classes and $\mathcal{Y}_{\mathrm{unseen}} \subset \mathcal{Y}$ be the unseen gesture classes, with $\mathcal{Y}_{\mathrm{in}} \cap \mathcal{Y}_{\mathrm{unseen}} = \emptyset$. For subset $S \in \{\mathrm{tr}, \mathrm{val}, \mathrm{test}\}$,

$$\mathcal{D}_S^{\mathrm{in}} = \{(\mathbf{x}, \mathbf{p}, y) \in \mathcal{D}_S : y \in \mathcal{Y}_{\mathrm{in}}\}, \qquad \mathcal{D}_S^{\mathrm{unseen}} = \{(\mathbf{x}, \mathbf{p}, y) \in \mathcal{D}_S : y \in \mathcal{Y}_{\mathrm{unseen}}\}.$$

***Definition 2. Unimodal Encoders.*** Let $\mathcal{E}_x$ and $\mathcal{E}_p$ map EMG and pose, respectively, into $\mathbb{R}^d$. When the pose encoder is used frozen, we write $\mathcal{E}_p^*$.

**Unimodal Encoder Pre-training.** We adopt a Transformer encoder (Vaswani et al., 2023) with a linear tokenizer to map raw signals into $d$-dimensional token embeddings. A patch length $S$ along time yields $L = \lfloor T/S \rfloor$ non-overlapping tokens. We flatten channels within each patch and project to $\mathbf{a}_i \in \mathbb{R}^d$. Following masked autoencoders (MAE) (He et al., 2021), we randomly mask a certain ratio of input tokens. The encoder processes only unmasked tokens and the transformer decoder reconstructs all tokens. The reconstruction loss is a mean squared error loss only applied to the masked tokens. We pre-train the EMG and pose encoders independently, yielding $\mathcal{E}_x$ and $\mathcal{E}_p$. We use a mask ratio of $0.5$ and a patch length of $S = 200$. Unlike CLIP, which is trained on billions of image–text pairs, we align strong unimodal encoders to reduce the need for large-scale paired data.

### 2.2 EMBRIDGE

The proposed cross-modal representation learning framework EMBridge comprises three components: (i) a Querying Transformer that acts as an information bottleneck, extracting pose-informative features from the EMG encoder; (ii) a Masked Pose Reconstruction Loss that strengthens representation learning; and (iii) a Community-Aware Soft Contrastive Learning objective that aligns the relative geometry of the EMG and pose spaces by matching their community-level similarity structures, yielding a more structured EMG latent space. The framework is shown in Figure 1.

### 2.2.1 QUERYING TRANSFORMER (Q-FORMER).

Inspired by BLIP-2 (Li et al., 2023), we use a set of learnable queries to extract pose-informative features from the EMG encoder. Let $Q^{(0)} \in \mathbb{R}^{M \times d}$ be $M$ learnable queries. The Q-Former $F_\phi$ stacks 4 self-attention blocks (each with a feed-forward layer) and 2 cross-attention layers inserted every other block, whose keys/values are from $\mathcal{E}_x(\mathbf{x})$. The self-attention modules of the Q-Former are initialized from the pre-trained Pose-MAE encoder $\mathcal{E}_p^*$ while cross-attention layers are randomly initialized. All Q-Former parameters and the EMG encoder $\mathcal{E}_x$ are trainable, while the pose encoder $\mathcal{E}_p^*$ remains frozen. The Q-Former takes the learnable queries $Q^{(0)}$ as input and produces the same length of queries with learned representations $Q' \in \mathbb{R}^{M \times d}$. We train the Q-Former using a contrastive objective following Li et al. (2022). Given the pose embedding $v_i = \mathcal{E}_p^*(\mathbf{p}_i)$, we select the query token that has the highest cosine similarity with $v_i$ as $u_i = Q'_{m^*(i)}, m^*(i) = \arg\max_{m \in \{1,\dots,M\}} \frac{Q'^\top_m v_i}{\|Q'_m\| \|v_i\|}$. Given a mini-batch of size $B$, we define the softmax over EMG–pose similarities $q_{ij} = \frac{\exp(u_i^\top v_j / \tau)}{\sum_{k=1}^B \exp(u_i^\top v_k / \tau)}$. Let $I \in \{0, 1\}^{B \times B}$ be the one-hot indicator matrix with $I_{ij} = 1$ iff $j = i$ (matching pairs on the diagonal). The Information Noise-Contrastive Estimation (InfoNCE) loss (van den Oord et al., 2019) can then be written as

$$\mathcal{L}_{\text{InfoNCE}} = -\frac{1}{B} \sum_{i=1}^B \sum_{j=1}^B I_{ij} \log q_{ij}. \tag{1}$$

Unlike CLIP, where both encoders are trained jointly, in our setup the pose encoder is frozen and serves as a fixed anchor, since pose representations are higher-quality and more structured compared to EMG representations. Thus, we adopt the standard InfoNCE loss (EMG→pose) rather than a symmetric variant, since gradients only flow into the EMG encoder while the pose embeddings remain fixed. By encouraging the learnable queries to extract EMG features that are most consistent with pose representations, the Q-Former efficiently and effectively aligns EMG and pose.

### 2.2.2 MASKED POSE RECONSTRUCTION LOSS (MPRL)

Beyond the contrastive loss, we add a masked reconstruction objective on input pose tokens to enrich the representation learning process (Li et al., 2023). Specifically, in the first forward pass, the Q-Former produces query embeddings $Q' \in \mathbb{R}^{M \times d}$. In the second forward pass, we concatenate masked pose tokens $\tilde{P} = [\tilde{p}_1, \dots, \tilde{p}_L]$ with $Q'$ and apply the same self–attention modules with an attention mask: pose tokens may attend to all queries, while queries attend only to themselves. Cross–attention from pose tokens to EMG is also disabled, pose cannot directly access EMG features. Thus, the information required to reconstruct masked pose tokens must be captured in $Q'$, enforcing that queries extract pose-informative content from EMG. Let $\mathcal{M}$ be the set of masked pose-token indices and $H_P = F_\phi([Q; \tilde{P}]) \in \mathbb{R}^{(M+L) \times d}$ denote the outputs of Q-Former. The mask ratio is $r$. A decoding layer $g$ maps the outputs back to input pose token space. We minimize

$$\mathcal{L}_{\text{MPRL}} = \frac{1}{|\mathcal{M}|} \sum_{m \in \mathcal{M}} \left\| g(H_P[m]) - P[m] \right\|_2^2, \tag{2}$$

where $P[m]$ is the ground-truth pose token at index $m$. Jointly optimizing $\mathcal{L}_{\text{MPRL}}$ with the contrastive loss encourages the Q-Former to learn denoised, pose-informative EMG representations and then yields an EMG latent space with richer pose semantics that can extrapolate to new gestures.

### 2.2.3 COMMUNITY-AWARE SOFT CONTRASTIVE LEARNING (CASCLE)

We propose CASCLe, which aligns EMG to pose by matching relative geometry of embedding spaces rather than only instance-level pairs. In standard contrastive learning, all non-matching poses are treated as negatives, even if some are semantically very close to the true positive. This treatment is suboptimal because grouping similar poses as strict negatives confuses the model and leads to unstable gradients. CASCLe addresses this by assigning soft targets. Poses that are more similar to the ground-truth pose receive higher probabilities, while relatively dissimilar ones receive lower or zero probabilities. Since the pose encoder $\mathcal{E}_p^*$ is frozen, its embedding space defines a fixed relational graph. CASCLe builds soft targets from this graph and trains the Q-Former queries to learn similar

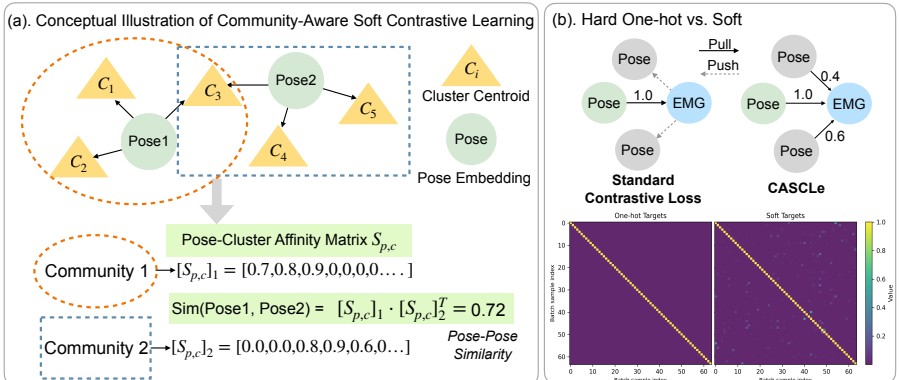

Figure 2: Unlike conventional contrastive loss that relies on one-hot targets, (a). CASCLe constructs soft targets based on community-level similarity. Each community is represented by affinities to cluster centroids, and pose–pose similarity is computed from affinity vectors. Soft targets used in CASCLe are shown in (b), computed from a batch of 64 samples for clearer visualization.

neighborhood structures for EMG embeddings, which strengthens overall semantic consistency with pose. An illustration of CASCLe is shown in Figure 2.

*Pose communities.* We cluster pre-trained Pose-MAE embeddings (offline) using $k$-means (Likas et al., 2003) to obtain $N_c$ centroids $\mathcal{C} \in \mathbb{R}^{N_c \times d}$ (all $\ell_2$-normalized). For a mini-batch of pose embeddings $P = [v_1; \ldots; v_B] \in \mathbb{R}^{B \times d}$, we compute a *pose–cluster affinity matrix* $S_{p,c} = P\mathcal{C}^\top \in \mathbb{R}^{B \times N_c}$, where each row is sparsified by keeping only the top-$k_c$ closest clusters. This keeps the community size reasonable and excludes irrelevant clusters, reducing noise in soft targets.

$$[S_{p,c}]_{ij} \leftarrow \begin{cases} [S_{p,c}]_{ij}, & j \in \mathrm{TopK}(S_{p,c}[i,:]), \\ 0, & \text{otherwise.} \end{cases} \tag{3}$$

*Pose–pose similarity matrix.* The community-aware pose–pose similarity matrix is then defined as $S_{p,p} = S_{p,c} S_{p,c}^\top \in \mathbb{R}^{B \times B}$. To prevent self-matches from dominating the probability distribution after softmax when generating soft targets, we remove the diagonal: $\bar{S}_{p,p} = S_{p,p} - \mathrm{diag}(S_{p,p})$. A similar strategy has been adopted in prior work (Gao et al., 2023) for soft target construction and proven effective. *Soft targets* are then defined as

$$\tilde{y}_{ij} = \frac{\exp\big(\bar{S}_{p,p}[i,j]/\tau_s\big)}{\sum_{k \neq i} \exp\big(\bar{S}_{p,p}[i,k]/\tau_s\big)}, \quad j \neq i, \tag{4}$$

with temperature $\tau_s > 0$. Intuitively, $\tilde{y}_{ij}$ is the probability that pose $v_j$ is a semantically relevant neighbor of $v_i$ in the fixed pose relational graph. Using the same EMG-pose similarities $q_{ij}$ defined earlier for InfoNCE, CASCLe minimizes a *Soft contrastive objective* defined as the cross-entropy between soft targets $\tilde{y}_{ij}$ and $q_{ij}, i \neq j$:

$$\mathcal{L}_{\mathrm{CASCLe}} = -\frac{1}{B} \sum_{i=1}^{B} \sum_{j=1}^{B} \tilde{y}_{ij} \log q_{ij}. \tag{5}$$

This objective can be interpreted as predicting the degree of similarity between EMG–pose pairs, where this degree is measured according to the structural organization of the pose latent space. The total training objective of EMBridge combines instance-level and structural community-level supervision:

$$\mathcal{L} = \mathcal{L}_{\mathrm{InfoNCE}} + \alpha\,\mathcal{L}_{\mathrm{CASCLe}} + \lambda\,\mathcal{L}_{\mathrm{MPRL}},$$

with weights $\lambda, \alpha > 0$. InfoNCE enforces instance alignment, while CASCLe aligns relational structure of the latent space between two modalities. In this way, the EMG encoder is guided to align not only with its exact pair but also with poses that share similar semantics, providing more robust and informative supervisory signals.

Table 1: Dataset splits with gesture and user counts. Four unseen gestures evaluated out of six total.

| Split (totals) | Subset | Gesture Counts | | User Counts | |
|---|---|---|---|---|---|
| | | In-dist. | Unseen | In-dist. | Unseen |
| $\mathcal{D}_{tr}$ (23 gestures / 158 users) | $\mathcal{D}_{\text{probe-tr}}^{\text{in}}$ | 4 | 0 | 158 | 0 |
| $\mathcal{D}_{val}$ (29 gestures / 15 users) | $\mathcal{D}_{\text{tune}}$ | 4 | 4 | 0 | 3 |
| | $\mathcal{D}_{\text{probe-tr}}^{\text{unseen}}$ | 0 | 4 | 0 | 12 |
| $\mathcal{D}_{test}$ (29 gestures / 158 users) | $\mathcal{D}_{\text{eval}}$ | 4 | 4 | 0 | 20 |

## 3 EXPERIMENTS AND RESULTS

### 3.1 EVALUATION PROTOCOLS.

We employ two evaluation protocols to examine the learned representation quality. **Linear probing** (LP): with labeled EMG data $\{(x_i, y_i)\}$, we freeze $\mathcal{E}_x$ and train a randomly-initialized linear classifier $\mathcal{C}$ on top of its embeddings, reporting accuracy on a held-out split. **Zero-shot classification** (ZS) is performed as k-nearest-neighbor voting in the embedding space, following standard practice in representation learning (Marks et al., 2024; Radford et al., 2021). For each EMG sample, we retrieve its top-$k$ nearest poses in the embedding space, then vote the corresponding gesture labels to determine the predicted gesture. Given a test EMG sample $x_j$, let $\mathcal{R}_j = \text{TopK}_{p \in \mathcal{P}_{\text{test}}} \big( \mathcal{E}_x(x_j)^\top \mathcal{E}_p^*(p) \big)$ be the set of $k$ pose samples with highest cosine similarity to $x_j$. We then predict $\hat{y}_j = \text{mode}\big\{ y(p) \mid p \in \mathcal{R}_j \big\}$, where $y(p)$ is the gesture class of pose $p$.

### 3.2 DATASETS

***emg2pose* dataset.** We use *emg2pose* (Salter et al., 2024), a large-scale open-source EMG dataset containing 370 hours of sEMG and synchronized hand pose data across 193 consenting users, 29 different behavioral groups that include a diverse range of discrete and continuous hand motions such as making a fist or counting to five. The hand pose labels are generated using a high-resolution motion capture system. The full dataset contains over 80 million pose labels and is of similar scale to the largest computer vision equivalents. Each user completed four recording sessions per gesture category, each with a different EMG-band placement. Each session lasted 45–120 s, during which users repeatedly performed a mix of 3–5 similar gestures or unconstrained freeform movements. We use non-overlapping 2-second windows as input sequences. EMG is instance-normalized, band-pass filtered (2–250 Hz), and notch-filtered at 60 Hz. For more details, please refer to Salter et al. (2024).

**Data Split for *emg2pose*.** We evaluate on two disjoint gesture sets drawn from the public *emg2pose* corpus. First, we select four representative single-hand motions covering various finger movements as our **in-distribution gestures**. Second, from the six held-out classes that are not seen during training, we exclude the two-handed gesture and the highly variable "finger freeform" class, yielding four **unseen gestures**. Details of gesture classes are in Appendix. For data splits, we follow the public train $\mathcal{D}_{tr}$, val $\mathcal{D}_{val}$, test $\mathcal{D}_{test}$ splits and define our data splits for downstream gesture classification tasks as shown in Table 1. The model is pre-trained on the full $\mathcal{D}_{tr}$. A linear head is trained on $\mathcal{D}_{\text{probe-tr}}^{\text{in}}$, and report final accuracy on the evaluation set $\mathcal{D}_{\text{eval}}^{\text{in}}$. For linear probing on unseen gestures (which appear only in the original val and test splits), we train on $\mathcal{D}_{\text{probe-tr}}^{\text{unseen}}$, and report accuracy on $\mathcal{D}_{\text{eval}}^{\text{unseen}}$. Zero-shot classification is evaluated only on $\mathcal{D}_{\text{eval}}$. All users in $\mathcal{D}_{\text{eval}}$ are unseen, so both the LP and ZS results also assess user-level generalization. A held-out dataset $\mathcal{D}_{\text{tune}}$, strictly disjoint from all other sets, is reserved for hyper-parameter tuning.

***NinaPro* dataset:** We utilized two *NinaPro* EMG datasets for a more comprehensive evaluation of EMBridge. Specifically, Ninapro DB2 (Atzori et al., 2014) is used for pre-training , which includes paired EMG-pose data from 40 subjects. It contains 49 hand gestures (including basic finger flexions, functional grasps, and combined movements) performed by 40 healthy subjects. EMG signals are recorded from 12 electrodes placed on the forearm at a sampling rate of 2 kHz, alongside hand kinematics data captured by a data glove. For downstream gesture classification, we use NinaPro DB7 (Krasoulis et al., 2017), which contains data from 20 non-amputated subjects collected with

the same EMG device and gesture set as DB2 (more details on NinaPro Website[1]). **Data split.** The entire DB2 dataset was used for pre-training, except Gestures 1, 5, 10, and 15 from exercise B, which were excluded to serve as **unseen gestures** in DB7. Gestures 1, 5, and 10 from exercise C were used as **in-distribution gestures**. Within each gesture, sessions were randomly divided into probe-training and evaluation sets, and zero-shot evaluation was conducted only on the latter.

### 3.3 GESTURE CLASSIFICATION RESULTS

**Comparing Schemes.** We evaluate EMBridge against various baselines. Unimodal models trained solely on EMG include: a supervised encoder–decoder Transformer (**PoseT**) regressing poses from EMG; the supervised regression models from the *emg2pose* benchmark (Salter et al., 2024) (**emg2pose**, **Vemg2pose**, **NeuroPose**); and a self-supervised MAE model trained only on EMG (**EMG-MAE**). We also compare to multi-modal models: a CLIP-style Contrastive Pose–EMG Pre-training framework (**CPEP**) (Cui et al., 2025), which applies $\mathcal{L}_{\text{InfoNCE}}$ directly to [CLS] tokens from EMG and pose encoders via a projection layer; and a plain **Q-Former** variant trained only with $\mathcal{L}_{\text{InfoNCE}}$. Unlike Q-Former, CPEP does not introduce a transformer but uses a projection layer. We further evaluate label-smoothed variants of both models. Label smoothing has been shown to improve contrastive learning by mitigating overconfidence and handling noisy similarities (Wickstrøm et al., 2022; Li et al., 2022). We introduce **CPEP-LS** and **Q-Former-LS**, where InfoNCE targets are softened with a smoothing factor of $0.1$. For a fair comparison under linear probing, each baseline's encoder is frozen and a softmax linear head is trained on top. As an additional reference, we report an **upper bound** from linear probing the pre-trained Pose-MAE. Since the pose encoder is the fixed alignment anchor, this represents the best achievable performance if EMG features were perfectly aligned. For LP, we use publicly available *emg2pose* checkpoints with the same data splits.

We evaluate EMG representation quality on both **in-dist.** and **unseen** gestures using two protocols: zero-shot classification (ZS) and linear probing (LP). Supervised baselines do not support zero-shot (K-nearest neighbor) classification in the embedding space. We report balanced accuracy on both *emg2pose* and *NinaPro* dataset to account for class imbalance across gesture classes. As shown in Table 2, EMBridge consistently outperforms all baselines, with the largest gains in zero-shot classification, where it even surpasses the LP performance of all unimodal models. The most significant improvements appear on unseen gestures for *emg2pose* and ZS on in-dist. gestures for *NinaPro*, demonstrating the stronger generalization capacity of EMBridge and its practical value for wearable gesture recognition. We note that CPEP achieves higher LP performance on unseen gestures than EMBridge for both datasets. This is likely due to our use of query averaging in EMBridge instead of selecting the query with maximum similarity to the paired pose, which may be suboptimal. Maximum-similarity selection is avoided here during LP to prevent potential data leakage. We also evaluate EMBridge in the few-shot setting with LP on *emg2pose*. Even with only 50% of the probe-training data, EMBridge outperforms all unimodal baselines.[2]

**Per-Gesture Classification Performance Breakdown.** We conduct a detailed analysis on the *emg2pose* dataset by computing per-gesture F1 scores from zero-shot classification on unseen gestures to assess performance gains achieved by EMBridge over other cross-modal frameworks (CPEP and Q-Former). Reporting F1 scores provides a more comprehensive and balanced view of per-class performance. Confusion matrices in Figure 3(a) offer a clearer illustration of improvements. Compared to CPEP and Q-Former, **EMBridge achieves consistently higher F1 scores across all gesture classes**, with particularly notable gains on *Class 1* (0.513 vs. 0.439/0.494) and *Class 3* (0.504 vs. 0.436/0.458). *Class 3* is the gesture of counting up and down then finger wiggling, which is very challenging for vision-based gesture classification due to visual occlusion (Salter et al., 2024). The improvements underscore EMBridge's stronger discriminative capability on difficult and frequently confused gestures, demonstrating its practical value in real-world cases where occlusion is common.

**Per-User Performance Gains Breakdown.** Similarly, on unseen gestures, we compute the zero-shot classification performance (F1 score) within each user, where all 20 users are held out from training. Figure 3(b) illustrates the per-user performance gains of EMBridge compared to CPEP and Q-Former. For per-user analysis, EMBridge achieves an overall improvement of $14.2\%$ over CPEP (0.522 vs. 0.457) and $10.2\%$ over Q-Former (0.522 vs. 0.473). The average relative per-user im-

---

[1] https://ninapro.hevs.ch/
[2] More details are provided in the Appendix.

Table 2: Comparison of gesture classification results across unimodal and multi-modal models. Results are reported on the *emg2pose* dataset and the *NinaPro* dataset.

| Unimodal Models | *emg2pose* | | *NinaPro* | |
|---|---|---|---|---|
| | In-dist. LP | Unseen LP | In-dist. LP | Unseen LP |
| Upper-bound | 0.851 | 0.649 | 0.769 | 0.632 |
| EMG-MAE | 0.347 | 0.334 | 0.283 | 0.256 |
| NeuroPose (Salter et al., 2024) | 0.692 | 0.248 | / | / |
| emg2pose (Salter et al., 2024) | 0.734 | 0.405 | / | / |
| Vemg2pose (Salter et al., 2024) | 0.650 | 0.312 | / | / |
| PoseT | 0.705 | 0.433 | 0.694 | 0.425 |

| Multi-modal Models | In-dist. | | Unseen | | In-dist. | | Unseen | |
|---|---|---|---|---|---|---|---|---|
| | LP | ZS | LP | ZS | LP | ZS | LP | ZS |
| CPEP | 0.782 | 0.757 | 0.536 | 0.481 | 0.675 | 0.604 | 0.483 | 0.413 |
| CPEP-LS | 0.780 | 0.759 | **0.538** | 0.487 | 0.681 | 0.617 | **0.494** | 0.424 |
| Q-Former | 0.782 | 0.763 | 0.493 | 0.498 | 0.688 | 0.613 | 0.481 | 0.447 |
| Q-Former-LS | 0.777 | 0.760 | 0.495 | 0.498 | 0.692 | 0.618 | 0.486 | 0.439 |
| **EMBridge** | **0.785** | **0.777** | 0.505 | **0.528** | **0.703** | **0.692** | 0.492 | **0.447** |

provement is $16.0\%$ compared to CPEP and $11.6\%$ compared to Q-Former, which demonstrates that EMBridge yields consistent improvements across unseen users, even under inter-subject variability.

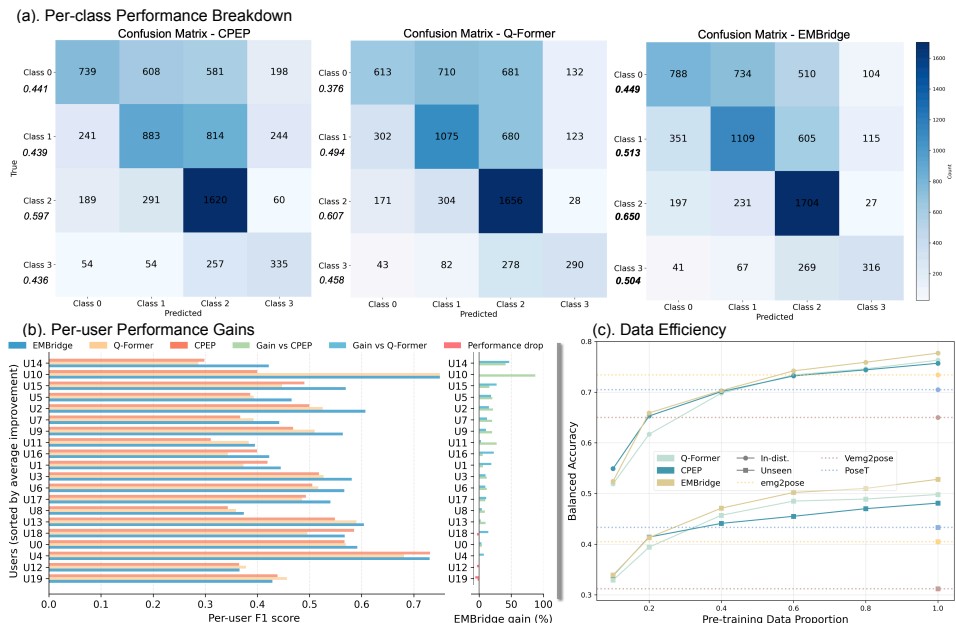

Figure 3: (a) Confusion matrices from ZS on unseen gestures, with per-class *F1 scores* shown beside row labels. (b) Per-user ZS performance on unseen gestures. (c) Data efficiency analysis via ZS on in-dist. and unseen gestures. Dotted lines indicate LP performance of unimodal baselines.

## 4    ABLATION STUDY

**Individual Contribution of Components in EMBridge.** We analyze three ablated variants of EM-Bridge, each removing a component (Q-Former, MPRL, or CASCLe) to assess its individual impact. Removing Q-Former reduces the model to CPEP + CASCLe, since without the Q-Former architecture and learnable queries, the masked pose reconstruction task cannot be performed. As shown in

Table 3: Ablation of EMBridge: individual component impact and soft contrastive objectives.

| Ablated Variants | LP in-dist. | LP unseen | ZS in-dist. | ZS unseen |
|---|---|---|---|---|
| EMBridge w/o Q-Former | 0.793 | 0.538 | 0.763 | 0.494 |
| EMBridge w/o MPRL | 0.783 | 0.494 | 0.764 | 0.516 |
| EMBridge w/o CASCLe | 0.784 | 0.485 | 0.764 | 0.509 |
| Soft Contrastive Objective | | | | |
| Label Smoothing (Fini et al., 2023) | 0.777 | 0.489 | 0.759 | 0.511 |
| SoftCLIP (Gao et al., 2023) | **0.788** | 0.490 | 0.760 | 0.510 |
| CASCLe | 0.785 | **0.505** | **0.777** | **0.528** |

Table 3, removing any component leads to a drop in zero-shot performance. Interestingly, removing Q-Former yields slightly better linear probing results, consistent with prior CPEP findings, and demonstrates the versatility of CASCLe, which can be effectively integrated into the CPEP architecture to improve performance. Removing MPRL or CASCLe reduces generalization to unseen gestures, underscoring their importance in cross-modal alignment and representation learning.

**Soft Contrastive Objectives.** We further compare CASCLe with alternative soft contrastive learning objectives. **Label Smoothing**, explored in Fini et al. (2023), applies soft targets in CLIP and has shown consistent gains. We also adapt SoftCLIP (Gao et al., 2023) to our EMG–pose setup by deriving soft targets from instance-level pairwise similarities between pose samples, providing a fair baseline against CASCLe, which models community-level structural similarity. We replace $\mathcal{L}_{\text{InfoNCE}} + \alpha \mathcal{L}_{\text{CASCLe}}$ with label smoothing and the adapted SoftCLIP objective, respectively, while keeping the rest of EMBridge unchanged. Further discussion of soft contrastive learning is provided in Section 5. In Table 3, CASCLe outperforms both alternatives in ZS, which highlights the advantage of modeling community structure over simple soft labels or instance-level similarities.

**Data Efficiency.** We investigate how the scale of paired pre-training data influences downstream performance. Since collecting paired EMG–pose data is costly and time-consuming, it is essential to evaluate the data efficiency of EMBridge. We uniformly downsample sessions within each gesture class to simulate limited pre-training data. We vary the proportion of paired pre-training data from 0.2 to 1.0 (full dataset) and report ZS performance on both in-dist. and unseen gestures in Figure 3(c). Remarkably, even when trained with only $40\%$ of the paired data, EMBridge's zero-shot classification still surpasses the LP performance of unimodal baselines trained on the full dataset.

**Sensitivity to Hyper-parameters.** We analyze how ZS and LP performance of EMBridge is influenced by different hyper-parameter choices. Specifically, we vary one hyper-parameter at a time while keeping all others fixed to their optimal settings. Results are summarized in Figure 4. The choice of $\tau_s$ and the number of top-$k_c$ clusters (used in CASCLe; see Eq. 3) jointly determines the quality of the resulting soft targets. A smaller $\tau_s$ produces a sharper probability distribution, and as $\tau_s \to 0$, CASCLe degenerates to the Supervised Contrastive Loss (Khosla et al., 2021) by assigning a hard label of 1 to multiple positives. In contrast, a moderate $\tau_s$ assigns reasonable weights to off-diagonal soft positives, balancing contributions without allowing them to dominate or amplifying small differences. Increasing the number of top-$k$ clusters ($k = 10, 20$) degrades both LP and ZS performance (Figure 4b), confirming that including distant neighbors introduces noise when constructing soft targets. EMBridge is robust to variations in $\lambda$ and mask ratio $r$ in the masked pose reconstruction task, as well as loss weight $\alpha$. We also find that using 16 queries yields the best results. 8 queries may be insufficient to capture the full range of information, while 32 queries can lead to overfitting. We also evaluate alternative similarity metrics for CASCLe. By default, cosine similarity is used to compute both the pose–cluster and pose–pose similarity matrices. Using L1 and L2 distances yields weaker results. This aligns with prior contrastive learning literature, where cosine similarity is commonly preferred for representation alignment (Radford et al., 2021; Chen et al., 2020). In Figure 4 (f), we observe that using more nearest neighbors improves ZS performance.

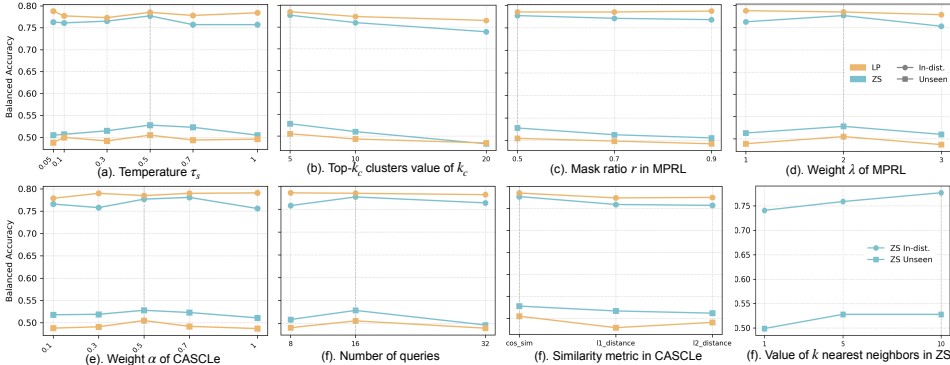

Figure 4: Sensitivity to hyper-parameters. Dashed lines indicate the values used in the best setup.

## 5  DISCUSSION AND CONCLUSION

**Related Work on Soft Contrastive Learning.** Traditional contrastive learning methods (van den Oord et al., 2019; Chen et al., 2020; Radford et al., 2021) use one-hot targets, where only the exact matching pair is treated as positive and all others as negatives. However, when multiple positives or highly similar instances exist, this introduces false negatives. Recent work addresses this by incorporating soft targets. Fini et al. (2023) employs label smoothing to generate soft targets. SoftCLIP (Gao et al., 2023) derives them from intra-modal similarity using fine-grained image features, while X-CLR (Sobal et al., 2024) builds a sample similarity graph and replaces binary labels with similarity scores. SoftCLT (Lee et al., 2024b) softens targets based on temporal proximity, and Huang et al. (2024) aligns cross-modal and uni-modal representations using teacher-derived similarity signals. **CASCLe** differs by constructing soft targets from **community-level** structural similarities in the embedding space, rather than relying solely on instance-level relations.

The motivation of the asymmetric setup is to use pose representations as fixed anchors to guide the learning of EMG representations. Training both encoders simultaneously could lead to suboptimal representations, as the pose representations are altered to be closer to the noisier EMG representations. Additionally, this asymmetric setup enhances extendability. Future work can pre-train the pose encoder on large-scale and unpaired pose data using MAE. This allows us to effectively learn EMG representations by aligning with frozen pose embeddings, even when paired data is limited. Leveraging abundant unpaired data reduces reliance on paired samples and improves data efficiency.

By leveraging pose as a rich supervisory signal, EMBridge learns pose-informed EMG embeddings that capture structural and semantic relationships and are discriminative in the latent space. Across in-distribution and unseen gesture classification tasks, EMBridge demonstrated strong performance gains, particularly in the zero-shot setting. Overall, EMBridge provides an effective approach to zero-shot gesture classification on wearable EMG signals, and can serve as a foundation for future exploration of cross-modal representation learning on EMG and other bio-signals.

## 6  LIMITATIONS AND FUTURE WORK

While EMBridge significantly enhances zero-shot gesture classification on wearable EMG signals, we acknowledge limitations and room for future exploration. Our framework currently relies on paired EMG-pose data for cross-modal alignment. Although *emg2pose* and *NinaPro* are large-scale datasets, high-quality paired datasets remain scarce in the broader bio-signal community. Training on a single dataset could potentially limit the pose encoder's capacity. A promising direction for future work is to leverage large-scale, publicly available unpaired pose data for unimodal pre-training. This would yield more robust pose representations, providing stronger supervision for EMG representation learning while significantly reducing the reliance on paired samples. EMBridge can further demonstrate its effectiveness in cross-modal learning with limited paired data. Furthermore, the proposed framework can potentially be extended to other modalities such as RGB-EMG or Video-EMG in future work, which can be done by incorporating a pre-trained vision encoder. Another interesting

direction for future work is to explore probabilistic modeling when constructing pose communities. Specifically, a Gaussian Mixture Model could be used to assign each pose embedding a soft probability distribution over multiple clusters or communities. Structural similarity between poses could then be computed based on the similarity between these membership distributions (e.g., using KL divergence), enabling a smoother and more continuous modeling of pose neighborhood structure.

## 7    REPRODUCIBILITY STATEMENT

All datasets used in this work are publicly available. Details of the model architecture, training objectives, and implementation details are provided in Section 2 and Appendix A.6, including all key hyper-parameters and model configurations to ensure reproducibility.

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

# A    APPENDIX

## A.1    THE USE OF LARGE LANGUAGE MODELS (LLMS)

We used LLMs to assist with language polishing, including grammar checking and word choice refinement. All research ideas, experiment design, analyses, and essential contributions were conducted by the authors.

## A.2    DETAILS OF EMG2POSE DATASET

### A.2.1    DATA COLLECTION PROTOCOL

Data collection was divided into four distinct sessions, involving multiple device don-doff cycles (avg. 3.9) to account for sensor placement shifts. In each session, participants performed two repetitions of prompted stages (gesture categories), with each stage lasting 45 to 60 seconds. Annotations are available at this recording level, which we utilized as the gesture class labels. Each stage contains either a mix of discrete gestures or freeform unprompted movements. To ensure a broad range of postures, participants were explicitly instructed to move their hand from right-to-left and up-and-down while performing specific gestures.

### A.2.2    DETAILS OF GESTURES

**In-distribution (4 classes).**    $\mathcal{D}_{\text{eval}}^{\text{in}}$ contains approximately 80 sessions per class from 20 held-out users.

*class 0:*  Thumb swipes whole hand; *class 1:*  Hand claw, grasp, and flicks. *class 2:*  ThumbsUp-Down, ThumbRotations; *class 3:*  FingerPinches, SingleFinger, PinchesMultiple;

**Unseen (4 classes).**    $\mathcal{D}_{\text{eval}}^{\text{unseen}}$ uses the same 20 held-out users, with approximately 80 sessions each for *classes 0–2* and 40 sessions for *class 3*.

*class 0:*  HookEmHorns, OK, and Scissors; *class 1:*  Shaka and Vulcan peace; *class 2:*  Counting up/down face side away; *class 3:*  Counting up/down with finger wiggling and spreading.

## A.3    VISUALIZATION OF REPRESENTATIONS.

We visualize EMG embeddings before and after applying EMBridge using t-SNE (Van der Maaten & Hinton, 2008), and compare them with anchor pose embeddings from the pre-trained Pose-MAE. Points are colored by gesture classes. Before EMBridge, EMG embeddings (pre-trained with MAE), show mixed distributions across classes. After EMBridge, the EMG embeddings become more structured and separable across classes. As we observe in the pose space, some overlap between classes remains, which reflects micro-gestures within 2-second windows that share semantic similarity across gesture categories. This highlights both the improved alignment achieved by EMBridge and the intrinsic difficulty of gesture classification on this dataset.

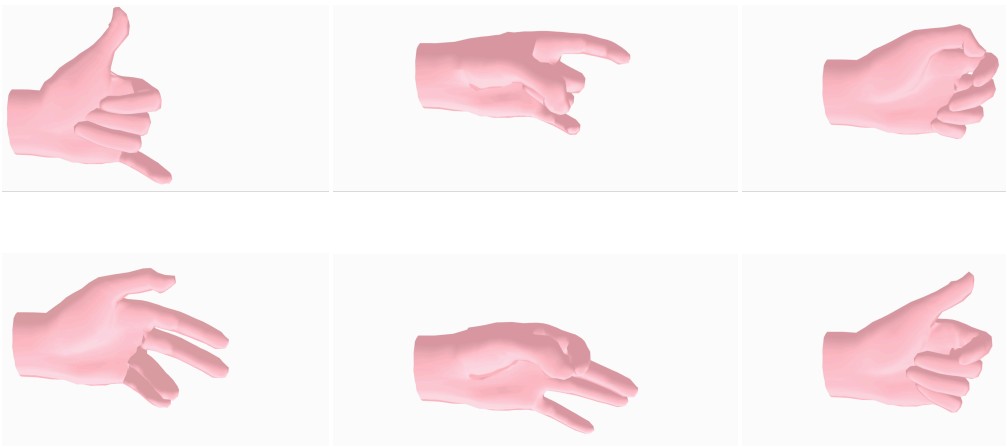

Figure 5: Example visualizations of gestures used in gesture classification tasks.

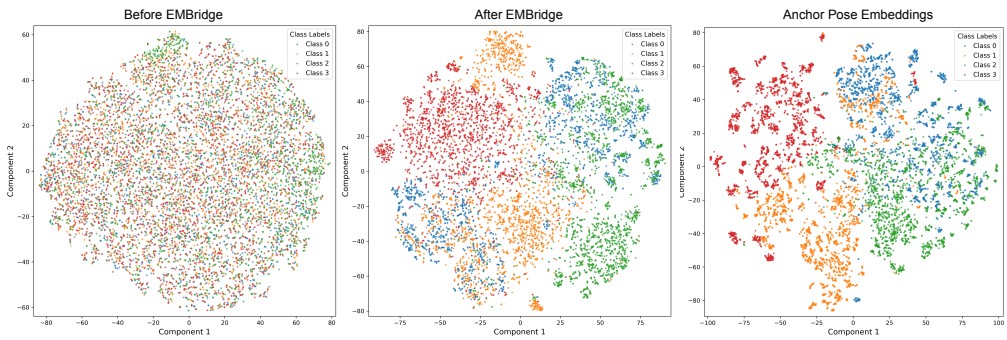

Figure 6: t-SNE visualization of embeddings from in-dist. gestures, colored by gesture class labels.

### A.4 FEW-SHOT EVALUATION OF EMBRIDGE

We evaluate the few-shot performance of EMBridge by gradually increasing the number of training samples within each class (n-shot) during linear probing. For each number of shots, we repeat random sampling five times to obtain a more reliable estimate of performance. We report the average balanced accuracy, with the standard deviation indicated as a shaded region in Figure 8. With only 50% of the probe-training data (40 shots), EMBridge can achieve performance almost comparable to that of the full set on unseen gestures and surpasses all baselines trained on the complete dataset.

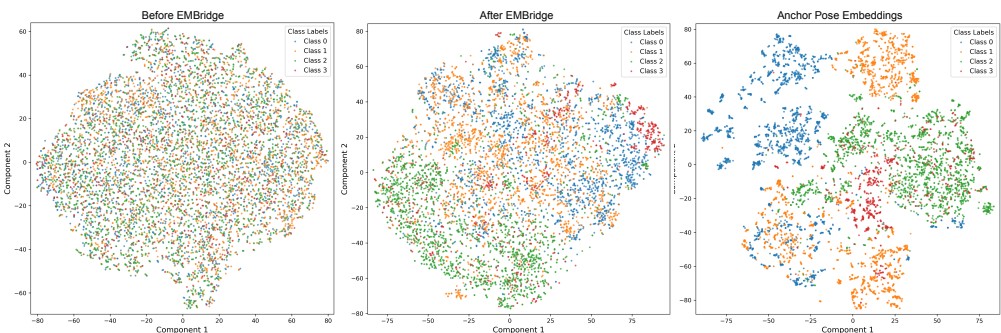

Figure 7: t-SNE visualization of embeddings from unseen gestures, colored by gesture class labels.

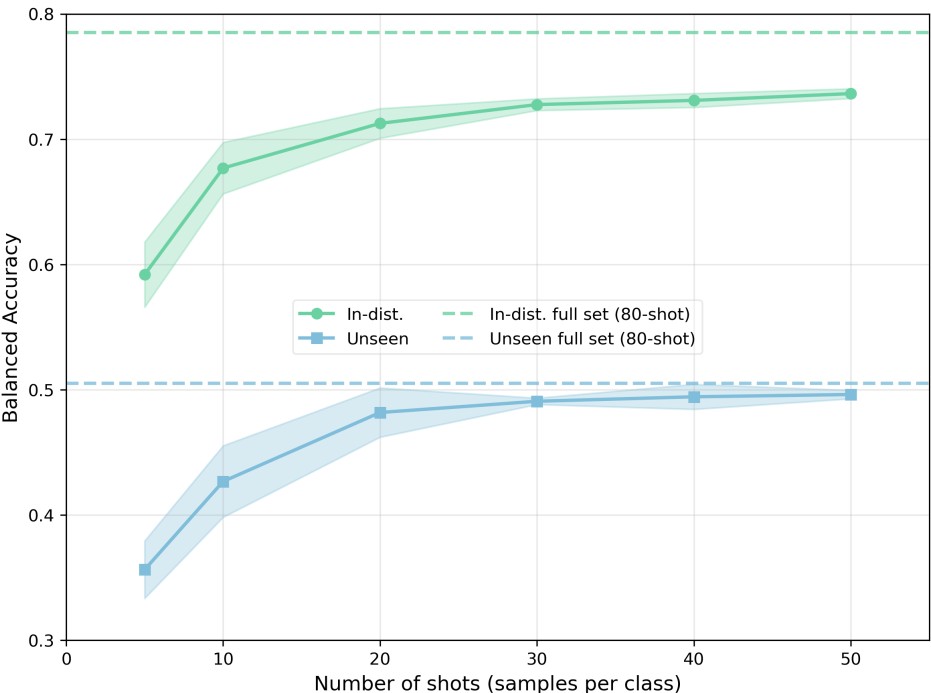

Figure 8: Few-shot evaluation of EMBridge. X-axis is the number of training samples within each class (n-shot) during linear probing. For each number of shots, we repeat random sampling five times to obtain a more reliable estimate of performance. We report the average balanced accuracy, with the standard deviation indicated as a shaded region.

### A.5    VISUALIZATION OF SOFT TARGETS CONSTRUCTED IN CASCLE

We visualize the soft targets to examine how the temperature $\tau_s$ and the number of top-$k$ clusters $k_c$ influence the resulting probability distribution. As shown in Figure 9, a smaller $\tau_s$ produces a sharper distribution, where probability mass is concentrated on a few dominant samples. This indicates that the model places high confidence on a small number of nearest pose neighbors, approaching a hard-label regime. In contrast, a larger $\tau_s$ leads to a smoother distribution with more evenly distributed weights, reflecting greater uncertainty and incorporating information from a broader set of pose samples. For the number of clusters $k_c$, larger values expand the set of contributing clusters and potentially diluting the impact of the most similar samples.

### A.6    IMPLEMENTATION DETAILS

We use 2 s windows sampled at 2 kHz for both pose and EMG. EMG is instance-normalized, band-pass filtered (2–250 Hz), and notch-filtered at 60 Hz. Following (Salter et al., 2024), we apply channel-rotation augmentation to EMG. Our MAE is an encoder–decoder Transformer model with 4 encoder layers and 2 decoder layers, and the embedding dimension is $d=256$. We optimize with AdamW (Loshchilov & Hutter, 2019) (lr $4e-4$, weight decay $1e-5$) and cosine annealing with warm restarts (Loshchilov & Hutter, 2017). Token length is $S=200$ for pose and $S=50$ for EMG, producing non-overlapping tokens along time. Mask ratio $r = 0.5$. Each MAE is trained for 100 epochs. PoseT is an encoder-decoder transformer model that consists of 4 encoder layers and 2 decoder layers, trained using the same losses adopted in Salter et al. (2024).

For CPEP, we attach a 1-layer projection head (hidden size 256) to the EMG encoder and train the EMG encoder plus projection head while keeping the pose encoder frozen. More details about CPEP can be found in Cui et al. (2025). The contrastive temperature $\tau$ is learnable (initialized to 0.02). All output embeddings are $\ell_2$-normalized. All model trainings are conducted on $4\times$ NVIDIA V100 GPUs; end-to-end training of each model takes approximately 6 to 8 hours. CPEP, Q-Former, and

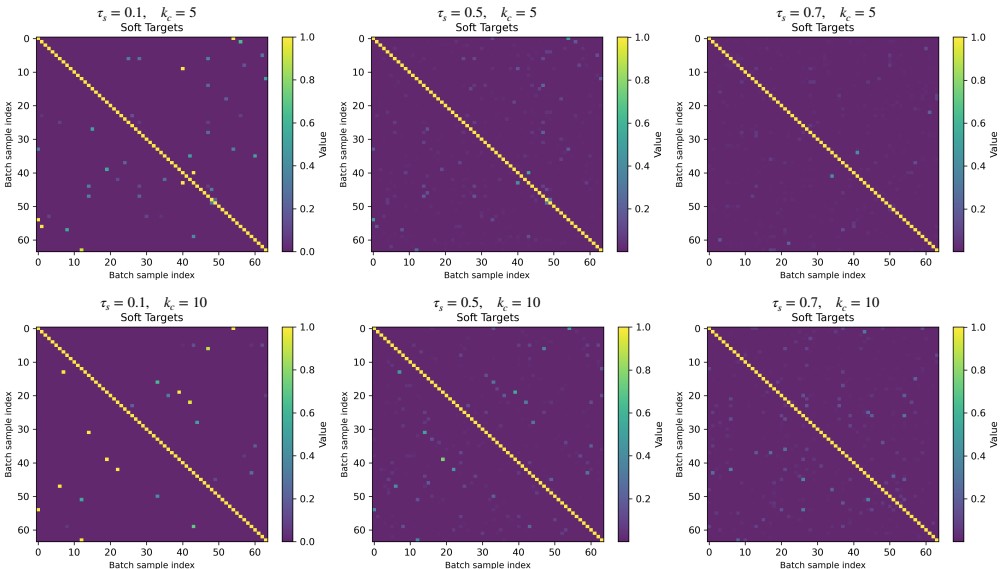

Figure 9: Soft targets from a batch of 64 samples for clearer visualization. We vary the value of temperature $\tau_s$ and the number of top-k clusters $k_c$.

EMBridge are all trained for 40 epochs. $N_c = 138$ clusters were computed using K-means from the anchor pose embeddings. The output embedding size used in linear probing is 256. Batch size is 256 for all models. Increasing batch size (512, 1024) for CPEP and Q-Former will degrade the performance, but EMBridge's performance remains robust as shown in Table 4.

Table 4: Impact of batch size on zero-shot classification performance of unseen gestures.

| Method | Batch Size 256 | Batch Size 512 | Batch Size 1024 |
|---|---|---|---|
| CPEP | 0.481 | 0.439 | 0.433 |
| Q-Former | 0.498 | 0.464 | 0.436 |
| **EMBridge (Ours)** | **0.528** | **0.525** | **0.523** |

In zero-shot retrieval, we precompute pose embeddings for the entire corpus and, for each EMG query, retrieve the top-$k$ neighbors by cosine similarity with $k=10$. The predicted label is the majority vote of the retrieved labels. For linear probing on *emg2pose* benchmark models, we replace the final decoding layer with a randomly initialized linear layer for classification, we average the embeddings at each timepoint as input to the classification layer.

