# OpenReview forum: "EMBridge: Enhancing Gesture Generalization from EMG Signals Through Cross-modal Representation Learning"
_ICLR.cc/2026/Conference — ICLR 2026 Poster_

### Official Review · Reviewer_fYtB · 2025-10-17

**Soundness:** 3
**Presentation:** 3
**Contribution:** 2
**Rating:** 6
**Confidence:** 4

**Summary:**

This paper presents a cross-modal framework that aligns EMG signals with hand pose embeddings to enable zero-shot gesture recognition. The approach is technically solid and yields consistent improvements.

**Strengths:**

1.The proposed solution represents a well executed and effective integration of recent state-of-the-art multimodal representation learning approaches, such as BLIP-style Q-Formers, into the EMG–pose alignment setting. The framework adapts these techniques in a way that is coherent and practically meaningful for wearable gesture recognition.
2.The experimental results are comprehensive and show consistent improvements across multiple datasets and evaluation protocols, including both in-distribution and zero-shot gesture classification, demonstrating the practical effectiveness of the proposed method.

**Weaknesses:**

1.While the proposed Community-Aware Soft Contrastive Learning objective introduces a soft alignment mechanism, the community structure itself is derived from a hard K-means clustering, which inherently imposes discrete partitions on the pose embedding space. This feels contradicting the goal of modeling continuous semantic similarity. Should a probabilistic model such as a Gaussian Mixture or kernel-based affinity estimation better capture soft neighborhood structures?
2.The construction of communities purely from geometric proximity in the pose latent space overlooks available semantic priors about gesture types. Integrating gesture semantics (e.g., coarse action categories) or using supervised/semi-supervised clustering could produce more meaningful communities and stronger cross-modal supervision.
3.While the paper is well structured and the experimental settings are clearly described, the background context for this specific application domain remains somewhat underdeveloped. In particular, the work would benefit from a clearer definition and motivation of the pose modality (e.g., what exactly constitutes a “pose” sample, how gestures are defined, and what practical application scenarios are targeted). For such a specialized domain, providing stronger domain background and contextual grounding would make the contribution easier to interpret and appreciate.

**Questions:**

NA

---

> ### Author Response · Authors · 2025-11-21
>
> We sincerely thank the reviewer for acknowledging the effectiveness of our framework. We are deeply grateful for the thorough review and constructive comments, which have greatly helped us with improving the quality and presentation of our manuscript. We address the specific concerns below.
>
> ---
>
> *W1: Hard K-means Clustering*
>
> **Response:**
>
> We appreciate the insights and valuable suggestion.
> While K-means partitions the space discretely, CASCLe utilizes these partitions softly. We do not use hard cluster assignments as targets. Instead, we use a vector of distances to multiple centroids to represent the position of a pose sample in the embedding space, and determine the similarity between paired poses by calculating the cosine similarity of these vectors. We then generate a probability distribution over the pose-to-pose similarities as soft targets (Eq. 4). This allows the model to capture soft semantic structures.
>  However, we acknowledge the reviewer's insight about using probabilitic models to capture soft neighborhood structures.  A Gaussian Mixture Model could assign each pose embedding a soft probability distribution over multiple clusters or communities. Similarity between two poses could then be computed based on the similarity of their membership distributions (e.g., via KL divergence), potentially enabling a smoother and more continuous pose neighborhood structure.
>  We included this as potential future work in the revised paper (Section 6).
>
> ---
>
> *W2: Why Geometric proximity, not gesture semantics?*
>
> **Response:**
>
> We appreciate this insightful comment. The reviewer proposed a very interesting idea and could be valuable for other datasets with more fine-grained gesture labels. Because the emg2pose dataset only provides a very coarse labeling of gesture categories (detailed description in Appendix A.2.1), which can consists of a mix of 2-3 gestures, using gesture labels as semantic priors can be misleading.
> For example, the gesture "Counting 3" is  much more similar to the "OK Sign"  than it is to "Counting 1", though counting 1 and counting 3 belong to the same gesture category. So using semantics like gesture labels to define communities would sometimes mislead the model to group dissimilar poses together while separating similar ones.
>
> So we relied on geometric proximity in the pose embedding space rather than high-level labels (like "Counting") to define communities. Furthermore,
> using supervised labels (e.g., gesture classes) to define communities could bias the model toward specific gesture categories in the training set. Leveraging the geometry of the embedding space allows the model to capture kinematic components and micro-gestures. This granular representation enhances zero-shot generalization on unseen gestures. Because although unseen gestures are categorically new, they are composed of familiar kinematic components already encoded by the model.
>
> ---
>
> *W3: Domain Background and Contextual Grounding*
>
> **Response:**
>
> We greatly appreciate this helpful suggestion.
> We clarified the definitions of hand poses and gestures in Section 2.1. We also emphasized the motivation for using pose modality, domain background and targeted practical applications in Introduction paragraph 1 and 2.
> Specifically, we explicitly define:
> * **Pose:** Hand skeleton represented by 20 joint angles at a certain timepoint.
> * **Gesture:** A temporal trajectory of poses over a window $T$ corresponding to a specific category.
>
> Pose data directly captures the kinematic structure of hand movements, making it an informative modality for guiding EMG representation learning.
>
> The potential applications of this work can be wearable Human-Computer Interaction in scenarios such as VR/AR and prosthetics. In these scenarios, a wrist-worn device must continuously infer hand gestures from EMG signals to drive a virtual avatar or robotic hand.
>
> - **The Challenge**: A critical bottleneck in this domain is that users cannot be expected to record training data for every possible movement they might perform.
>
> - **Our Contribution**: Our framework enables zero-shot generalization, allowing the system to recognize novel gestures without requiring the user to provide training samples for them.

---

> > ### Comment · Reviewer_fYtB · 2025-11-22
> > **Thank you for the rebuttal**
> >
> > I thank the authors for providing the rebuttal. I think the paper is technically sound, without clear flaws, and above the acceptance threshold of ICLR. I do not have further questions.

---

### Official Review · Reviewer_jAkZ · 2025-10-21

**Soundness:** 3
**Presentation:** 3
**Contribution:** 3
**Rating:** 6
**Confidence:** 3

**Summary:**

This paper investigates the task of hand gesture recognition. The authors propose a cross-modal representation learning framework that bridges the semantic gap between EMG representations and poses to achieve zero-shot gesture generalization. Specifically, they introduce a Q-Former, a masked pose reconstruction loss, and a community-aware soft contrastive learning objective to enhance EMG representation learning. Extensive experiments are conducted to validate the effectiveness of the proposed framework.

**Strengths:**

1. The proposed cross-modal representation learning method for hand gesture recognition is a novel and interesting attempt.

2. The proposed cross-modal representation learning strategy effectively improves the quality of EMG representations.

3. The authors proposed a community-level structural similarity framework for soft contrastive learning.

4. The authors performed zero-shot classification on both in-distribution and unseen gesture categories.

**Weaknesses:**

1. The proposed community-aware soft contrastive learning mainly stems from previous work on contrastive learning. However, the motivation for introducing community-level structural similarities is not well explained.

2. The proposed method is only validated on the pose–EMG paired data. Could the proposed method also be applied to RGB–EMG data?

**Questions:**

1. In Appendix A.6, the authors claim that batch size does not affect the proposed EMBridge, while it degrades the performance of CPEP and Q-Former. Do these conclusions come from experimental results?

2. The evaluation is conducted on the emg2pose and NinaPro datasets. Only four gestures are used in the unseen setting, would this setup reduce the difficulty of the zero-shot evaluation?

---

> ### Author Response · Authors · 2025-11-21
>
> We thank the reviewer for recognizing the novelty of our cross-modal strategy and the effectiveness of our approach. We appreciate the constructive feedback and detailed questions, which are very helpful for further improving our paper. Below, we address the specific concerns raised.
>
> ---
>
> *W1: The motivation for introducing community-level structural similarities is not well explained.*
>
> **Response:**
> We appreciate the opportunity to clarify our motivation. The motivation for CASCLe arises from the continuous nature of pose data, which standard contrastive learning fails to capture.
>
> * **Limitation of Standard Contrastive Learning:** A gesture is composed of multiple pose components (e.g., finger flexions, finger spreading) that can be shared across different gestures. Standard contrastive learning (like InfoNCE) treats all non-matching samples as equally distant negatives. For hand gestures, this is suboptimal because different gesture classes can share semantic similarities of certain 2-second window pose samples. For example, *Counting the Number 3* is structurally closer to *OK Sign* than to *Counting the Number 1*. This can be shown in Appendix A.3 Figure 6 and 7, where we visualize the pose embedding space using t-SNE. We observe that some pose samples from different gesture classes are closely clustered.
>
> * **CASCLe** We introduce communities to preserve this structural similarity. By pre-clustering the high-quality pose space, we identify communities of kinematic similarity. CASCLe uses these communities to assign soft targets, guiding the EMG encoder to learn a structured representation space that brings EMG embeddings corresponding to similar poses closer.
>
> We have integrated clarified motivation into the Introduction (Paragraph 3) of the revised manuscript
>
> ---
>
> *W2:  Could the proposed method also be applied to RGB–EMG data?*
>
> **Response:**
> Yes, the EMBridge framework is generic and can be extended to RGB-EMG data.
> The pose encoder in our framework can be replaced with a pre-trained vision encoder (e.g., ViT or ResNet).
>  We added this as future work in Section 6.
>
> ---
>
> *Q1: Robustness to batch size supported by experiment results?*
>
> **Response:**
> Yes, the conclusion are supported by our experimental results. We have added the results of different batch sizes in Table 4 of Appendix A.6.
>
> | Method | Batch Size 256  | Batch Size 512 | Batch Size 1024 |
> | :--- | :---: | :---: | :---: |
> | **CPEP** | 0.481 | 0.413 | 0.316 |
> | **Q-Former** | 0.498 | 0.431 | 0.328 |
> | **EMBridge** | **0.528** | **0.526** | **0.523** |
>
> Table: Impact of batch size on zero-shot classification performance of unseen gestures using emg2pose dataset.
>
> ---
>
> *Q2: Only four gestures are used in the unseen setting, would this setup reduce the difficulty of the zero-shot evaluation?*
>
> **Response:**
>
> We followed the data splits defined in the emg2pose paper (Salter et al., 2024) to ensure a fair and standardized comparison with emg2pose benchmark models. The held-out gestures were not selected by us but are fixed by the benchmark. We use 4 unseen gestures for NinaPro dataset to keep the classification tasks consistent across both datasets.
>
> These unseen classes are distinct and challenging: (1). They include dynamic, sequential movements (e.g., "Counting") rather than just static gestures. (2). They cover a wide variety of hand movements.
> The low classification accuracies of the baseline methods on unseen gestures (as shown in Table 2.) further confirm that this is a difficult, non-trivial zero-shot classification task.

---

> ### Comment · Reviewer_jAkZ · 2025-11-23
>
> Dear Authors,
>
> Thank you for your reply. My concerns have well adressed.
>
> Best regards.

---

### Official Review · Reviewer_VnDF · 2025-10-31

**Soundness:** 3
**Presentation:** 3
**Contribution:** 3
**Rating:** 8
**Confidence:** 4

**Summary:**

This paper focuses on gesture generalization from EMG signals and pose. The authors propose EMBridge, a cross-modal representation learning framework consisting of three components: (1) a querying transformer that extracts pose-informative features from EMG signals and aligns them with pose features using an asymmetric setup, (2) a masked pose reconstruction loss to enrich the pose representations, and (3) a community-aware soft contrastive learning objective to account for varying pose similarities that are not captured by standard contrastive learning. The authors conduct a detailed ablation study on different components of the framework, evaluate performance on both in-distribution and unseen gesture classification tasks, and demonstrate improvements over baseline methods.

**Strengths:**

The paper is well written, with clear motivation and challenges. It addresses an important and interesting problem in EMG-based gesture generalization to achieve zero-shot gesture classification. The proposed framework is original, well thought out, and carefully designed to meet domain-specific needs, such as incorporating pose communities for community-aware soft contrastive learning, which is particularly interesting. The experimental section is detailed and includes ablation studies on different components of the framework, along with analyses on three EMG datasets.

**Weaknesses:**

I do not see any major weaknesses in this work. However, a more detailed discussion of the limitations and potential future directions would strengthen the paper. While reproducibility details are sufficient, sharing the code in the future would further benefit the community.

**Questions:**

- What is the motivation behind using an asymmetric setup with the pose estimator and EMG encoder? Have the authors experimented with a setup where the pose estimator is also trained during the contrastive learning process (in the Query Transformer)?

- Regarding the relationship between paired gestures in experiments, are they independent or combinatorial? If they are combinatorial (for example, open hand + down), is this relationship captured in the pose similarity matrix, or could it be included as prior knowledge?

---

> ### Author Response · Authors · 2025-11-21
>
> We sincerely thank the reviewer for the positive and valuable feedback. We value your constructive suggestions, which have strengthened the revised manuscript.
> Below, we address your specific questions and suggestions.
>
> ---
>
> *W1: Limitations and future directions, sharing the code*
>
> **Response:**
>
> Thank you for the valuable suggestion. Our current study is limited by the availability of public paired datasets and this motivates a significant direction for future work: leveraging large-scale unpaired pose data for pre-training. By doing so, we aim to reduce the reliance on paired samples and demonstrate EMBridge's effectiveness even with limited paired EMG-Pose data. Another potential future work is to explore other modalities such as RGB-EMG or Video-EMG, which can be done by introducing a vision encoder.
> We added the detailed discussion of limitations and future work in Section 7 of the revised manuscript.
> We also plan to release the code upon paper publication.
>
> ---
>
> *Q1: Asymmetric Setup*
>
> **Response:**
>
> Thank you for raising this important question.
> We chose an asymmetric setup during training because the pose encoder produces higher-quality and more structured representations compared to the EMG encoder. This can be observed in the t-SNE visualization of pose and EMG embeddings in Figure 6 and 7. So the motivation is to use pose representations as a fixed anchor to guide the learning of EMG representations. Training both encoders simultaneously could lead to suboptimal representations, as the pose representations are altered to be closer to the noisier EMG representations.
>
> Additionally, this asymmetric setup enhances extendability. Future work can pre-train the pose encoder on large-scale and unpaired pose data using MAE. This allows us to effectively learn EMG representations by aligning with frozen pose embeddings, even when paired data is limited. Leveraging abundant unpaired data reduces reliance on paired samples and improves data efficiency.
> We clarified this design choice in Section 5 and Section 2.2.1.
>
> ---
>
> *Q2: Regarding the relationship between paired gestures in experiments, are they independent or combinatorial? If they are combinatorial (for example, open hand + down), is this relationship captured in the pose similarity matrix, or could it be included as prior knowledge?*
>
> **Response:**
>
> We appreciate this insightful question. The data collection protocol of emg2pose dataset inherently introduces combinatorial variations. In each 45–60 second session, participants performed specific gesture sequences while simultaneously moving their arm left-to-right and up-and-down. This ensures that the dataset captures specific hand shapes coupled with a diverse range of postures.
> We provide more details of the data collection process in Appendix A.2.1 for clarification.
>  In our current setup, the paired poses in the pose similarity matrix are treated as independent samples. Each embedding corresponds to a randomly sampled non-overlapping 2-second window, so each pose sample is unique. However, combinatorial relationships could be implicitly captured in the pose similarity matrix due to the nature of gestures collected. We do not explicitly model these combinatorial relationships in the current framework.
>  We thank the reviewer again for raising this interesting point.

---

> > ### Comment · Reviewer_VnDF · 2025-11-23
> >
> > Thank you for your response. My questions have been addressed, and I do not have any further questions.

---

### Meta-Review · Area_Chair_o6ay · 2026-01-02

**Summary:**

This work deals with gesture recognition from non-visual sources, EMG signals, which is addressed with a cross-modal representation. EMG signals are aligned with pose features and combined with masked reconstruction and contrastive learning.

The paper has received three positive reviews, with the reviewers appreciating a well-written paper and well-motivated work, which has been carefully executed and evaluated.

Minor weaknesses were the motivation of the asymmetric setup and the data generation in the experimental setup and possible alternative formulations. These weaknesses have been addressed in the rebuttal. All three reviewers acknowledged that their questions had been answered. A consensus was reached that this paper can be accepted.

The AC concurs (but is somewhat disappointed by the depth of the reviews and the discussions).

**Reviewer Concerns:**

See above.

**Reviewer Scores:**

This was apparent from the discussions, see above.

---

### Decision · Program_Chairs · 2026-01-26

Accept (Poster)